# Tissue Distribution of the Piscine Novirhabdovirus Genotype IVb in Muskellunge (*Esox masquinongy*)

**DOI:** 10.3390/ani12131624

**Published:** 2022-06-24

**Authors:** Robert K. Kim, Scott D. Fitzgerald, Matti Kiupel, Mohamed Faisal

**Affiliations:** 1Comparative Medicine and Integrative Biology, College of Veterinary Medicine, Michigan State University, 1129 Farm Lane, Room 340G, East Lansing, MI 48824, USA; path2fish@gmail.com; 2Department of Pathobiology and Diagnostic Investigation, College of Veterinary Medicine, Michigan State University, 4125 Beaumont Road, Building 0215, East Lansing, MI 48910, USA; fitzge18@msu.edu (S.D.F.); kiupel@dcpah.msu.edu (M.K.); 3Department of Fisheries and Wildlife, College of Agriculture and Natural Resources, Michigan State University, 1129 Farm Lane, Room 340G, East Lansing, MI 48824, USA

**Keywords:** viral hemorrhagic septicemia virus, pathogenesis, tropism, muskellunge, in situ hybridization

## Abstract

**Simple Summary:**

A novel strain of viral hemorrhagic septicemia virus was discovered in the Great Lakes. This Great Lakes strain of the virus infects a broad range of fish species with outcomes ranging from clinical and subclinical disease, persistent viral shedding, and/or death. Among the most susceptible species to the Great Lakes strain of the virus are juvenile muskellunge. Increased susceptibility to the virus in this regionally and economic important species generated a multitude of research questions to include but not limited to host range, pathogenesis, and diagnostic tools to efficiently detect and minimize viral spread into captive fish stocks. The overarching aim of the current study focuses on assessing the early and latter stages of disease progression in a battery of traditional and non-traditional diagnostically relevant tissues in juvenile muskellunge. Tissue damage from the virus and amount of live virus in each tissue were evaluated in conjunction with advanced diagnostic methods to identify cells targeted by the virus when possible.

**Abstract:**

A novel sublineage of the piscine novirhabdovirus (synonym: viral hemorrhagic septicemia virus), genotype IVb, emerged in the Laurentian Great Lakes, causing serious losses in resident fish species as early as 2003. Experimentally infected juvenile muskellunge (*Esox masquinongy*) were challenged with VHSV-IVb at high (1 × 10^5^ PFU mL^−1)^, medium (4 × 10^3^ PFU mL^−1^), and low (100 PFU mL^−1^) doses. Samples from spleen, kidneys, heart, liver, gills, pectoral fin, large intestine, and skin/muscle were collected simultaneously from four fish at each predetermined time point and processed for VHSV-IVb reisolaton on *Epitheliosum papulosum cyprini* cell lines and quantification by plaque assay. The earliest reisolation of VHSV-IVb occurred in one fish from pectoral fin samples at 24 h post-infection. By 6 days post-infection (dpi), all tissue types were positive for VHSV-IVb. Statistical analysis suggested that virus levels were highest in liver, heart, and skin/muscle samples. In contrast, the kidneys and spleen exhibited reduced probability of virus recovery. Virus distribution was further confirmed by an in situ hybridization assay using a VHSV-IVb specific riboprobe. Heart muscle fibers, hepatocytes, endothelia, smooth muscle cells, and fibroblast-like cells of the pectoral fin demonstrated riboprobe labeling, thus highlighting the broad cellular tropism of VHSV-IVb. Histopathologic lesions were observed in areas where the virus was visualized.

## 1. Introduction

Piscine novirhabdovirus (widely known as viral hemorrhagic septicemia virus, VHSV) is a World Organization of Animal Health (OIE) reportable pathogen that is noted for its ability to cause widespread mortality in a number of fish species throughout the northern hemisphere. VHSV belongs to the genus Novirhabdovirus, within the family *Rhabdoviridae* [1], and is historically known for its devastation in farmed rainbow trout from Europe [2,3]. The virus has four genotype (I–IV) and in 2003, a unique sublineage of the North American genotype (IV) emerged in the Great Lakes basin, designated VHSV-IVb [4]. The virus has been isolated and detected in >30 fish species in a geographical range that extends to all five Great Lakes and its presence is often associated with mass mortality episodes [5,6,7]. The virus has also been recovered from apparently healthy fish [5], amphipods [8], and piscicolid leeches [9]. In addition to its ecologic complexities, VHSV-IVb detection at known positive locations is inconsistent [10,11]. Although it is clear that VHSV-IVb persistence in the environment may depend on a number of factors, the ability of the virus to exist in hosts and go undetected in sample types other than the kidney, liver, and/or spleen by standard diagnostic methods remains unexplained and warrants further investigation.

Presently, direct detection of this OIE reportable virus relies upon protocols detailed in [12] and the American Fisheries Society Fish Health Section Bluebook [13]. These protocols recommend virus isolation on susceptible cell lines with confirmation through reverse transcriptase-polymerase chain reaction (RT-PCR). However, identification of VHSV using cell culture is dependent on a number of factors, including the tissue sample from which the isolation is attempted and the concentration of infectious virus particles present in the selected tissue sample [14]. Traditionally, most VHSV studies used kidneys and spleen samples for the initial isolation of VHSV. However, experimental studies have shown that different VHSV isolates vary in their tissue predilections depending on the particular strain of virus and species infected. For example, [15], who infected turbot (*Scophthalmus maximus*) experimentally with marine VHSV isolates, reported that infected turbot did not succumb to death in the early stages, nor did the extent of viral distribution in the turbot resemble that of the rainbow trout, (*Oncorhynchus mykiss*) experimentally infected with VHSV-Ia, as reported earlier by [16]. Using immunohistochemistry, the primary tissue targets of the virus in rainbow trout was the anterior kidneys and as early as 2 days post-infection (dpi), whereas in turbot the heart, kidney, spleen, and pancreas exhibited viral antigen as early as 7 dpi in the turbot. Such discrepancies can influence the choice of organs to be sampled for diagnosis.

The emerging VHSV-IVb sublineage exhibits a number of unique features that range from its fast substitutions in its nucleic acid [7], wide range of fish species, its exclusive presence in a freshwater system unlike most other VHSV strains, the plethora of lesions it causes, to the temperature range associated with wildfish mortality episodes, to variations in pathogenicity (reviewed in [11,17]). Variability in morbidity, mortality, and pathology is particularly apparent in VHSV-IVb, whose pathogenicity varied widely even among closely related fish species. VHSV-IVb exhibited low pathogenicity in steelhead trout (*Oncorhynchus mykiss*) and Chinook salmon (*Oncorhynchus tshawytscha*) to high pathogenicity in lake herring (*Coregonus artedi*), both of which are members of the family Salmonidae [18,19]. To this end, the primary objective of this study was to evaluate the viral distribution and load of VHSV-IVb strain MI03 in tissues of its highly susceptible host, the muskellunge (*Esox masquinongy*). The detection of viral nucleic acids through in situ hybridization (ISH) assays which utilize a specific nucleic acid probe has been applied to the detection of fish-pathogenic viruses to include other VHSV isolates [20,21]. The elevated sensitivity and specificity of this method can provide insight into the cellular predilections and sites where VHSV-IVb replicates in tissues of its susceptible host. The second objective of this study is to develop an in situ hybridization assay capable of detecting VHSV-IVb and then utilize the assay to assess the cellular targets of VHSV-IVb in the tissues of experimentally infected muskellunge. 

## 2. Materials and Methods

### 2.1. Cell Culture and Virus Propagation and Quantitation 

The Great Lakes VHSV genotype IVb index strain MI03, originally isolated in our laboratory in 2003 from muskellunge [4], was propagated in the *Epithelioma papulosum cyprini* (EPC) cell line [22]. Quantification of virus was accomplished through the viral plaque assay using EPC cells treated with polyethylene glycol and using methylcellulose overlay, as described in [23]. Cryogenic vials (Corning, Lowell, MA, USA) containing virus stock were kept at −80 °C until used. Serial dilutions were applied to the virus stock beginning with 1:100 to 1:10^12^. Plaques were allowed to form for a period of six days, and plaques were visualized by staining with crystal violet (0.5% *w*/*v*) following fixation with a 10% formalin solution buffered with phosphate buffered saline (PBS). The virus stock was calculated to have approximately 7.32 × 10^8^ plaque forming units/mL. 

Cell lines were maintained and subcultured in 150 cm^2^ tissue culture flasks (Corning, Lowell, MA, USA) at 25 °C using a growth formulation of Earle’s salt-based Minimal Essential Medium (MEM) (Invitrogen, Carlsbad, CA, USA) supplemented with 29.2 mg mL^−1^ L-glutamine (Invitrogen), Penicillin (100 IU mL^−1^) and Streptomycin (0.1 mg mL^−1^; Invitrogen), 10% fetal bovine serum (Hyclone, Logan, UT, USA), and sodium bicarbonate (7.5% *w*/*v*; Sigma-Aldrich, St. Louis, MO, USA).

### 2.2. Fish and Maintenance

Certified VHSV-free juvenile muskellunge (4 months post hatch) were procured from the Rathbun National Fish Hatchery (Moravia, IA, USA) and approved for use in experimental infection by Michigan State University’s Animal Care Program in accordance with the Institutional Animal Care and Use Committee (Michigan State University–Animal Care Program) protocol 07/07-123-00. Upon arrival, a subsample of five fish were euthanized using an overdose of tricaine methanesulfonate (MS-222; Argent Chemical Laboratories, Redmond, WA, USA) (25 mg mL^−1^) and tested for the presence of VHSV. Briefly, kidneys, heart, liver, and spleen were aseptically removed and homogenized using a Biomaster Stomacher (Wolf Laboratories Ltd, Pocklington, England.) at the high speed setting for 2 min. Homogenates were diluted with Earle’s salt-based minimal essential medium (MEM, Invitrogen, Waltham, MA, USA) supplemented with 12 mM tris buffer (Sigma-Aldrich), penicillin (100 IU mL^−1^), streptomycin (100 µg mL^−1^) (Invitrogen), and amphotericin B (250 μg mL^−1^) (Invitrogen) to produce 1:4 dilution (*w*/*v*) of original tissues. Samples were centrifuged at 2000× *g* and the supernatants inoculated into individual wells of a 24-well plate containing EPC cells grown with MEM (5% fetal bovine serum). Plates were incubated at 15 °C for 7 d and observed for the formation of cytopathic effects (CPE). A second blind passage was performed and assessed for the presence of CPE.

Fish were acclimated to temperatures from 16 °C to 11 °C over a three-week period before the experiment was initiated. All fish were initially held in a 1900 L circular fiberglass tanks in a continuous flow-through system supplied by oxygenated and facility chilled well water at Michigan State University-Research Containment Facility (East Lansing, MI, USA). Fish were sustained at approximate dissolved oxygen levels of 5 mg L^−1^ and water temperature of 11 °C at a flow rate of approximately 0.5–1 L min^−1^. The juvenile muskellunge were fed ad libitum with 2.0 mm sinking feed (Silver Cup, Murray, UT, USA) and transitioned to VHSV-free certified fathead minnows, *Pimephales promelas* purchased from Robinson Wholesale, Inc (Genoa City, WI, USA).

### 2.3. Experimental Infection of Juvenile Muskellunge

Experimental challenges were performed at the Michigan State University–Research Containment Facility (MSU-URCF, East Lansing, MI, USA). A group of 360 fish (weight 20.0 ± 10.9 g; total length 17.1 ± 1.5 cm) were randomly assigned into four groups. Groups of fish were challenged by immersion in water containing VHSV MI03 with Group 1 (90 fish) at a high dose 1 × 10^5^ PFU mL^−1^, Group 2 (90 fish) at a medium dose of 4 × 10^3^ PFU mL^−1^, Group 3 (90 fish) at a low dose of 100 PFU mL^−1^, and Group 4 (90 fish) served as a negative control by immersion in water mixed with 1 mL of sterile maintenance—MEM. All groups were held in static, oxygenated water for 90 min and subsequently divided into two tanks of equal numbers (45 fish). Calculated dose levels were in accordance to those described in [24].

### 2.4. Tissue Sampling

Samples of tissue were collected in parallel from four fish in each of the groups in fish infected at the medium dose, including the negative control. Moribund and/or apparently healthy fish were euthanized at predetermined hours (0 h, 6 h, 12 h, 24 h, 36 h) and days (2 d, 4 d, 6 d, 8 d, 15 d, 22 d, 29 d, 36 d, 43 d, 50 d, 57 d, 64 d) and subsequently dissected for collection of pectoral fin, gill, spleen, heart, liver, kidney, posterior intestine, and dorsal musculature with skin. Sampled tissues were divided into two groups, one of which would be stored at −80 °C in 1.5 mL centrifuge tubes (Denville Scientific, South Plainfield, NJ, USA) for the viral plaque assay and the second group to be fixed in phosphate buffered formalin (10%) for 24 h and embedded in paraffin wax for histopathology and in situ hybridization.

### 2.5. Viral Plaque Assay

Tissues were weighed and diluted with Earle’s salt-based minimal essential medium (MEM, Invitrogen) supplemented with 12 mM tris buffer (Sigma-Aldrich), penicillin (100 IU mL^−1^), streptomycin (100 µg mL^−1^) (Invitrogen), and amphotericin B (250 μg mL^−1^) (Invitrogen) at a range of 1:4 to 1:250 (*w*/*v*) and homogenized by mortar and pestle (Fisher Scientific, Pittsburgh, PA, USA). Homogenates were centrifuged at 2000× *g* and inoculated onto 24 well plates of confluent monolayers of EPC. Quantification of virus was completed in accordance with the methods described above. Plates were incubated at 15 °C for 6 d and stained to quantify the number of plaques in order to calculate the number of plaques per tissue.

### 2.6. VHSV Reisolation and Confirmation

Supernatant from tissue homogenates collected from VHSV challenged fish and negative controls were tested for the presence of VHSV using the EPC cell line. For PCR confirmation, total RNA was extracted using QIAamp Viral RNA Mini Kit (Qiagen, Valencia, CA, USA), according to the manufacturer’s instructions. Reverse transcription was accomplished by a two step protocol using the Affinity Script Multiple Temperature Reverse Transcriptase RT-PCR^TM^ (Stratagene, La Jolla, CA, USA) following the manufacturer’s instruction. The primer set used in this assay is recommended by [12] for detection of a 811 base pair sequence of the VHSV nucleocapsid gene: 5′-GGG-GAC-CCC-AGA-CTG-T-3′ (forward primer) and 5′-TCT-CTG-TCA-CCT-TGA-TCC-3′ (reverse primer). Polymerase chain reaction was achieved by adding into each reaction tube 5 μL of viral RNA, 50 pmol of each primer, 25 μL of GoTaq^®^ Green Mastermix DNA Polymerase (Promega), and the DNA-ase free water to create a final volume of 50 μL. The reverse transcriptase was inactivated by subjecting the mixture to 94 °C for 2 min, and 30 cycles of PCR (denaturation for 30 s at 94 °C, annealing for 30 s at 52 °C, and polymerization at 68 °C for 1 min) in a Mastercycler Personal Thermal Cycler (Eppendorf, Hauppauge, NY, USA). The polymerization was finalized by maintaining the mixture for a period of 7 min at 68 °C. The product was visualized by gel electrophoresis in 1.5% agarose gels. 

### 2.7. Histopathology

Once tissues were fixed in phosphate buffered formalin for 24–36 h, the tissues were trimmed and submitted for embedding into paraffin. Histologic sections were approximately 5 µm in thickness and stained with hematoxylin and eosin as detailed in [25]. 

### 2.8. In Situ Hybridization

Using the sequence of VHSV isolate MI03 (Gen Bank Sequence GQ385941.1), a riboprobe was created to anneal to a unique region of the nucleoprotein gene of VHSV-IVb at nucleotide position 279. The oligonucleotide probe sequence was thus 5′-GTC–CCA-TCA-TCT-TCG-CCA-AAG-CCA-CCA-A-3′. The probe was labeled with digoxigenin at the 5′ end (IDT). This specific probe was developed to differentiate VHSV-IVb from the other novirhabdovirus; IHNV, as well as all other VHSV genotypes. The use of the Basic Local Alignment Search Tool (www.ncbi.nlm.nih.gov/blast.cgi (accessed on 29 June 2011)) demonstrated no cross reactivity with IHNV, VHSV genotypes I-III, and VHSV-IVa. The probe was purified by high performance liquid chromatography (HPLC; IDT).

In order to maximize the sensitivity and specificity of this ISH assay, preliminary tests were performed in order to identify the optimal protocol and reagent concentrations as previously described [26]. ISH using the developed problem was tried on confluent monolayers of EPC cells in two separate 150 cm^2^ flasks that were infected with VHSV-IVb at approximately 6.6 × 10^2^ PFU/cm^2^. At 24 h post-infection, the cells of one flask were removed using a cell scraper (Corning) and subsequently pelleted by centrifugation at 200× *g* for 5 min. The supernatant was removed and immediately replaced with phosphate buffered formalin (10%). Removing EPC cells and fixation of the cell pellet was repeated by the steps described at 48 h post-infection. A negative control of EPC cells was obtained from an additional 150 cm^2^ flask in which the cells were not exposed to VHSV-IVb. 

Additionally, ISH using the developed probe was tried on tissues of infected muskellunge. Briefly, 5 μm thick sections were cut from paraffin-embedded tissues previously collected and placed onto positively charged slides, which were then deparaffinized and fixed using the Discovery XT automated slide-processing system (Ventana Medical Systems, Inc., Tucson, AZ, USA) as programed in the protocol for the RiboMap in situ hybridization reagent system (Ventana Medical Systems). Protease 3 (0.02 units mL^−1^ alkaline protease; Ventana Medical Systems) was used for 12 min at 37 °C for a proteolytic treatment followed by a mild cell conditioning step using the citrate buffer-based RiboCC reagent (Ventana Medical Systems) for 4 min at 95 °C. The slides were then denatured for 4 min at 37 °C, followed by hybridization for 1 h at 37 °C with the antisense oligonucleotide probe for VHSV-IVb suspended in hybridization buffer (RiboHybe; Ventana Medical Systems). The concentration used for the VHSV-IVb probe was 1.59 ng mL^−1^ (1:10,000 dilution). Four stringency washing steps were performed at 42 °C using 0.1× RiboWash (equivalent to 0.1× saline sodium citrate; Ventana Medical Systems) for 4 min for the first three and for 8 min for the fourth washing step. After the stringency washes, the slides were incubated with a rabbit monoclonal antidigoxigenin antibody (Invitrogen Corporation, Frederick, MD, USA) at a dilution of 1:10,000 for 32 min at 37 °C. Slides were then incubated in streptavidin-alkaline phosphatase conjugate (UMap anti-Rb AP; Ventana Medical Systems) for 16 min at 37 °C and the signal was detected automatically using the BlueMap nitroblue tetrazolium-BCIP (5-bromo-4-chloro-3-indolyl phosphate) substrate kit (Ventana Medical Systems) for 2 h at 37 °C. The final step involved counterstaining the slides with nuclear fast red-equivalent reagent Red Counterstain II (Ventana Medical Systems) for 4 min before adding a coverslip. Tissues collected from certified VHSV-IVb free naïve muskellunge were used as negative tissue controls whereas experimentally infected muskellunge with RT-PCR confirmed VHSV-IVb -positive tissues were used as positive controls. 

### 2.9. Statistics

Analysis of the data was completed using SAS Version 9.2 for Windows software package (SAS Institute Inc., Cary, NC, USA) in order to assess whether the organs/tissue could be differentiated and ranked based on the overall virus load measured. The dependent variable was denoted as the viral load whereas the independent variables were designated as the exposure dose level, sampling time, and organ/tissue sampled. Initially, the virus load obtained from each organ/tissue for individual fish was descriptively analyzed using PROC FREQ. Following a descriptive analysis of the data, logistic regression was applied to assess the probability of observing the virus in each organ/tissue for all fish.

Data were analyzed for virus loads greater than “0” followed by the log transformation of each value. Diagnostic evaluation of the data was completed using Tukey’s test of additivity in order to account for instances that may have only one virus load measured for a sampling time. The organs/tissues were compared to one another and ranked based on the paired t-test and through logistic regression, respectively. Subsequently, data analysis was performed using PROC GLM for a repeated measure 3-way ANOVA in instances where the virus load was attained in at least one of the eight sampled organs/tissues. 

## 3. Results

### 3.1. Virus Reisolation and Infectious Virus Load

Negative control fish tissues were free of virus by cell culture and RT-PCR at all sampling points. In the case of fish exposed to the high dose (1 × 10^5^ PFU mL^−1^), VHSV-IVb was not reisolated in any tissues collected between 0 h and 36 h p.i. (Table 1). However, VHSV-IVb was recovered as early as 2 d p.i. from the heart of two fish and the pectoral fin of one fish. By 6 d p.i. virus was isolated from all eight tissues as well as in subsequent sampling times, which revealed that the virus could be recovered from most tissues between 6 and 36 d p.i. At 43 d p.i., the virus was reisolated from the pectoral fin and skin/muscle samples in three of the four fish and the spleen of one fish. The number of virus isolations decreased further by 50 d p.i. where only one sample of the heart and pectoral fin were positive for VHSV-IVb in separate individuals. The virus was recovered from most tissues by 57 and 64 d p.i.

Of the total virus isolations from the sampled organs/tissues throughout the seventeen sampling time points, the heart, liver, and pectoral fin demonstrated the highest number of virus reisolations, which was a total of 16, followed by the skin/muscle at 14. The least recovery of VHSV-IVb was obtained from the spleen with a total of five and the kidney with eight. When the total number of isolations was evaluated for each sampling time, 6 and 15 d p.i appeared to have an increased number of virus recoveries. A total of 96 VHSV-IVb isolations were made of the 544 tissue samples collected. When infectious virus loads were evaluated in fish exposed at the high dose, the pectoral fin demonstrated the highest viral load with 6.63 × 10^7^ PFU g^−1^ at 6 d p.i., with levels decreasing in subsequent days. Interestingly, samples of pectoral fin overall also contained higher levels of virus as compared to the other tissues (Figure 1). Increased levels of VHSV-IVb were also found in heart samples at 15 d p.i. and periodically until 64 d p.i., reaching upwards of 4.26 × 10^7^ PFU g ^−1^. In all other tissues, levels of infectious virus were higher in fish sampled at 6, 8, and 15 d p.i.

In the case of fish exposed to the medium dose (4 × 10^3^ PFU mL^−1^), initial detection was limited to the pectoral fin of one fish as early as 24 h p.i. By 6 d p.i., VHSV-IVb was distributed to all sampled organs. However, by the following sampling time (8 d p.i.), virus recovery was limited to the skin/muscle of one fish. From 15 to 29 d p.i., VHSV-IVb was reisolated from nearly all tissue types, with a greater number of total isolations occurring at 29 d p.i. In subsequent sampling times, virus isolations decreased with the exception of 43 d p.i. in which a total of 18 virus isolations occurred. However, at 29 d p.i., a total of 22 virus isolations were observed from all tissue samples on that day. Similar to fish exposed at the highest dose, VHSV-IVb was recovered in 94 of the sampled tissues. By the end of the observation period, the total number of VHSV reisolations appeared to be more common in heart, skin/muscle, and gills, whereas it was less common in spleen samples. The titer of VHSV-IVb in the pectoral fin from one fish at 24 h p.i. was 9.75 × 10^3^ PFU g^−1^ tissue homogenate. From 6 to 29 d p.i., viral loads were consistently higher in all tissues that were confirmed VHSV positive (Figure 1). The highest viral load was detected in the heart of one fish at 6 d p.i. at 5.93 × 10^8^ PFU g^−1^, followed by the liver (5.20 × 10^8^ PFU g^−1^) and gill (5.04 × 10^8^ PFU g ^−1^) of the same fish (Table 2). The lowest amount of virus was in the skin and muscle tissue, which was 1.31 × 10^2^ PFU g ^−1^ at 8 d p.i. When assessing the viral load over the 64 d period, an overall decrease in viral load was noted in samples of the spleen, kidney, gill, and pectoral fin, whereas the heart, liver, posterior intestine, and skin/muscle exhibited fluctuating levels of virus among individual fish. 

In the case of fish infected at the low dose (100 PFU mL^−1^) with VHSV-IVb, there were overall many fewer virus reisolations which were limited to samples of heart, liver, intestine, gill, and pectoral fin (Table 3). The earliest recovery of virus occurred in the liver of one fish at 24 h p.i. and at 4 and 22 d p.i. Latter isolations in the intestine occurred in one fish sampled at each of the sampling times of 22, 29, and 43 d p.i. Single reisolations occurred in the heart at 36 h p.i., pectoral fin at 2 d p.i., and in the gill at 15 d p.i. The greatest number of virus reisolations occurred in both the liver and intestine of low dose infected fish. Although there were a limited number of virus reisolations in fish exposed at a low dose, the posterior intestine exhibited the highest viral load with 1.97 × 10^4^ PFU g^−1^ tissue followed by the pectoral fin with 4.47 × 10^3^ PFU g^−1^. In total, only nine isolations of VHSV-IVb occurred in low dose exposed fish.

### 3.2. Statistical Analysis

Comprehensive analysis of the viral load was completed through a frequency plot which provided an overview of instances in which the viral load could or could not be assessed. The overall percentage of when VHSV-IVb could be recovered from the tissues among the three exposure doses demonstrated that reisolation of virus was in general very low (Figure 2). The highest percentages were however found in the liver, heart, and skin/muscle tissue at 14.34%, 12.50%, and 11.76%, respectively. The lowest percentages of virus recovery were noted in the spleen and kidneys, which were 5.88% and 7.72%, respectively. Analysis by logistic regression and calculation of the odds ratio was determined by comparing the likelihood of recovering VHSV-IVb from each tissue versus the kidney. Analysis revealed that the chance of reisolating VHSV-IVb was highest in the liver (2.1917), followed by the heart (1.8267), and then in skin/muscle (1.6815) when compared to virus recovery from the kidney (1.000) (Table 4). Conversely, the lowest probability of reisolating VHSV-IVb was in the spleen and kidney (Table 4). 

When analysis of the mean infectious viral load was conducted by the paired *t*-test, results indicated that there was no significance in the mean viral load among the liver, heart, and skin/muscle samples, but it was significant when compared to the samples of gill, posterior intestine, kidneys, and spleen (Table 5). Furthermore, the mean infectious viral load of the pectoral fin was also significant as compared to the kidney and spleen (Table 5). 

### 3.3. Visualization of VHSV-IVb in Tissues of Infected Muskellunge

When the VHSV-IVb riboprobe was applied to 24- and 48-h post VHSV-IVb infection in EPC cells, positive staining was observed at both time points. In EPC cells that were not exposed to VHSV-IVb, no positive staining was observed.

Based on the infectious virus load in organs and tissues from muskellunge, histopathologic examination and ISH were completed on a representative number of samples. Histopathologic examination of gill sections revealed degenerative endothelial cells of the primary lamellae (Figure 3a). Following ISH, the same sight of the gills demonstrated intense riboprobe staining within endothelial cells (Figure 3a,b). Little to no change was observed in the secondary lamellae of the sampled sections. Pectoral fin sections demonstrated aggregates of pyknotic nuclei amongst fibroblast-like cells (Figure 4a), which also exhibited intense positive riboprobe staining in the same areas (Figure 4b). Additional lesions were also noted in the epithelial layers between fin rays. In the case of liver sections, degenerative and necrotic hepatocytes were observed from a number of fish (Figure 5a). Use of ISH revealed positive staining at the same locations (Figure 5b). In histologic sections of the heart, large foci of pyknotic and karyorrhectic nuclei of cardiac myocytes was frequently observed amidst aggregates of lymphocytes and macrophages (Figure 6a,b). Furthermore, riboprobe staining was most pronounced in these areas and more largely seen in cardiac myocytes (Figure 6c,d). 

Sections of kidney demonstrated cellular and karryorhectic debris scattered within the interstitium and tubular lumena (Figure 7a). Degenerate tubular epithelial cells were multifocally swollen and vacuolated (Figure 7a). Positive riboprobe labeling in the kidneys was noted in areas of necrosis (Figure 7b). 

Within the brain, aggregates of lymphocytic infiltrates were noted in the meninges and subjacent gray matter amidst areas of spongiosis and gliosis (Figure 8a). Riboprobe labeling of the brain was prominent in the meninges and largely absent in the gray matter (Figure 8b).

When sections of the posterior intestine were examined, aggregates of pyknotic nuclei were observed in the muscularis layers (Figure 9a), which also coincided with positive riboprobe staining at these locations (Figure 9b). In the case of skin and muscle sections, few to no changes were observed in the musculature and dermal layers (Figure 10a). However, the primary sights of riboprobe staining occurred in endothelial cells located at the junction of the dermis and underlying musculature (Figure 10b). Furthermore, positive staining was noted at the base of scales adjacent to compact layers of epidermis. The spleen of one fish sampled at 14 d p.i. was devoid of erythrocytes and lacked lymphoid tissue (not shown). However, all other spleen samples did not exhibit any pathologic changes and did not show any positive staining by ISH. No lesions or positive riboprobe signal were observed in histologic sections from the negative control group.

## 4. Discussion

The piscine novirhabdovirus (VHSV) is known for its wide host range infecting dozens of freshwater and marine fish species in the northern hemisphere [27]. Except for the rainbow trout (*Oncorhynchus mykiss*) and turbot, the distribution of each VHSV genotype and sublineage within its susceptible host(s) has not been thoroughly investigated because it was assumed that VHSV pathogenesis in other species resembles that of VHSV genotype Ia in the rainbow trout. This is particularly true for VHSV genotype IVb, the most recent VHSV sublineage to emerge [4] with its genome continues to undergo relatively fast substitutions [7]. Muskellunge is highly susceptible to VHSV-IVb [28] and is widely present in the Great Lakes basin, Upper Mississippi drainage, as well as inland lakes, rivers, and embayments in the Midwest [29] and therefore is an ideal candidate for surveillance and monitoring purposes in the waterbodies where VHSV-IVb is expected to spread. Shedding light on tissue distribution of the virus and its disease course in muskellunge deems necessary for the proper choice of samples for diagnostic purpose, thereby increasing the likelihood of detecting VHSV-IVb even if present in low titers. 

In the current study, after combining the mean infectious viral load in samples from all three dose levels, analyses revealed that the probability of detecting VHSV-IVb was highest in the liver, heart, and skin/muscle, in contrast to the kidneys and spleen in which VHSV-IVb was less likely to be recovered. These findings are surprising given that the kidneys and spleen are traditionally being sampled in most VHSV laboratory studies and field surveillance [30,31]. Although the value of using liver in VHSV-IVb detection has not been thoroughly explored, it has demonstrated its value in detection of other viral pathogens in fish such as infectious salmon anemia virus [32], golden shiner virus, koi herpes virus, and largemouth bass virus [33]. The use of liver homogenates in cell culture isolation has, at times, proven to be challenging given the resultant cytotoxicity that appears in the inoculated cell lines. 

In our study, heart samples also exhibited high loads of infectious virus from as early as 6 dpi and throughout the 64 d observation period for the medium and high dose exposure groups (Table 1 and Table 2). Brudeseth et al. [15], who studied tissue distribution of a number of VHSV marine isolates in the turbot following experimental infection, suggested that the VHSV isolates used in the study have a predilection for heart and kidney tissues as demonstrated by the strong positive signaling by immunohistochemistry and degenerative changes by histopathology. Unlike the kidneys, our study demonstrated that the heart of infected muskellunge has a relatively high virus load as well a strong, widely dispersed positive signals, making it an optimal organ to collect for diagnosis and with relative ease from the pericardial space, which are corroborated in previous studies with rainbow trout and alternate VHSV isolates [34]. Alternatively, in juvenile muskellunge challenged at a low dose, virus was reisolated from the intestines and liver on a total of three days of sampling as opposed to the heart, gill, and pectoral fin with only one day of virus recovery for each of the tissues. Although these findings suggest that the intestines could also be an ideal sample under instances of low virus exposure, virus recovery from only one fish at any time point is most likely incidental. More importantly, inclusion of intestine as a targeted tissue in virus reisolation would also prove challenging given the non-sterile environment of gastrointestinal contents that could pose a contamination risk to virology samples.

The relatively high virus load in skin and muscle samples and the strong positive signals in a number of cell types suggest a potential involvement of these tissues in the virus entry and shedding. Samples of muscle have previously demonstrated elevated levels of virus in the highly susceptible rainbow trout to VHSV-Ia [35]. Indeed, most of the ross lesions of VHSV are observed on the skin. Lovy et al. [36], who studied VHSV genotype IVa in Pacific herring (*Clupea pallasii*,) employing immunocytochemistry assay, suggested that early viral tropism is for the fish epidermal cells at 6 dpi followed by the virus spreading to the fibroblasts of the fin soft rays. In our experiment, detection of the virus in the pectoral fin was much earlier than what Lovy et al. [36] reported with the signals in pectoral fin fibroblast being very strong positive, indicating the VHSV-IVb is replicating at these sites. We believe that a clip of the pectoral fin should be included in the sample taken for diagnosis in the case of muskellunge. A similar notion has been reported in another fish-pathogenic rhabdovirus, the salmonid novirhabdovirus (formerly known as infectious hematopoietic necrosis virus, IHNV), by Harmache et al. [37], who used bioluminescence imaging to demonstrate that base of fins, particularly the pectoral fins, is the main portal of IHNV entry in experimentally infected rainbow trout. Isolation of VHSV-IVb from the intestine, together with its visualization of the virus within the intestinal wall by ISH, suggests that the gastrointestinal tract may play a role in virus shedding. The identification of virus in tissues of the intestinal wall, and pectoral fin by ISH, suggests that VHSV-IVb is replicating in these tissues and is not present due to contamination of tainted water or food pellets. Gills also emerged as a potential candidate for lethal and non-lethal sampling in the case of VHSV-IVb-infected muskellunge due to its relatively high virus load and strong ISH signals 6 dpi. On the contrary, Neukirch [38] was able to reisolate VHSV genotype Ia from the gills of experimentally infected rainbow trout as early as 1 d pi, which underscores the importance of determining disease course in each species because inherent differences in the virus strain, host species, and the detection methods play a major role in determining the disease course and hence the optimal choice of samples. 

The broad spectrum of cell types with positive ISH signals expresses the broad predilection of cellular targets by VHSV-IVb in muskellunge. The virus was visualized in cells originating from all three germ cell layers (mesoderm: renal tubules, smooth muscle of the gut, cardiac myocytes, and fibroblast; ectoderm: neuropil; endoderm: hepatocytes) indicating its pleiotropism. Additionally, histopathologic changes and riboprobe staining confirmed that the virus replicates in the fin, liver, and heart, resulting in damage to affected cells and tissues. Our findings in the muskellunge concur with those described by Kim et al. [39] for the olive flounder infected with VHSV-IVa in that as the infection advances, the virus becomes widely distributed in multiple organs, albeit at varying loads. Such information provides insights into the tissue/organ targeted for sampling in diagnostic assays. Results of virus load were supported by riboprobe staining which revealed the absence in the kidneys and spleen. This means that the virus loads in these two organs are below the detection limit of virus reisolation and ISH. Indeed, statistical analysis suggested that the kidneys and spleen were not ideal samples for VHSV-IVb recovery as compared to the liver and heart. 

## 5. Conclusions

The results of our study provide evidence for VHSV-IVb distribution in one susceptible host. The dynamics of virus–host interactions may vary from one host to another, a matter that requires additional studies with each of the species at risk. It is imperative that liver and heart be included in sampling for VHSV-IVb in the case of lethal sample testing. Additional processing steps to reduce the inherent cytotoxicity of the liver and environmental contaminants of the gill could enhance virus reisolation. Gills and fins, and the pectoral fin in particular, should be considered for the non-lethal sample scheme for VHSV-IVb screening.

## Figures and Tables

**Figure 1 animals-12-01624-f001:**
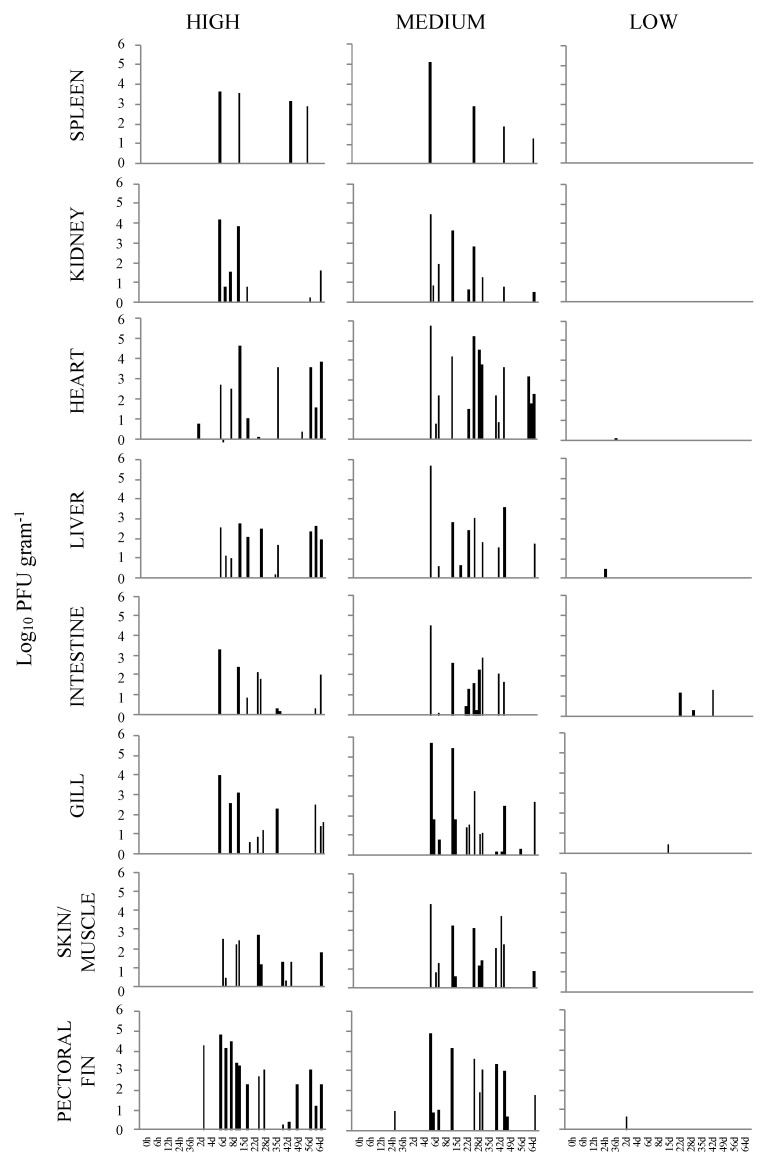
Load of viral hemorrhagic septicemia virus IVb (VHSV-IVb) determined by plaque assay, shown in log_10_ plaque forming units (PFU) gram^−1^ tissue in parallel tissue samples from four fish at predetermined time points. Fish were exposed VHSV-IVb concentrations of high (1 × 10^5^ PFU mL^−1^), medium (4 × 10^3^ PFU mL^−1^), and low (100 PFU mL^−1^).

**Figure 2 animals-12-01624-f002:**
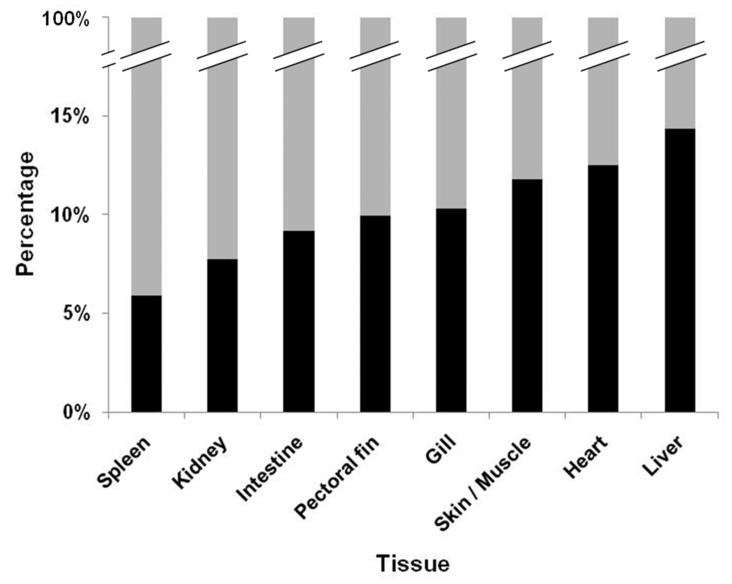
Combined total percentage of virus detections by viral plaque (black) among the high, medium, and low doses of infection. Non-detections are presented in gray. Concurrent, virus isolation by cell culture and confirmation of viral hemorrhagic septicemia virus IVb by reverse transcriptase-polymerase chain reaction was completed.

**Figure 3 animals-12-01624-f003:**
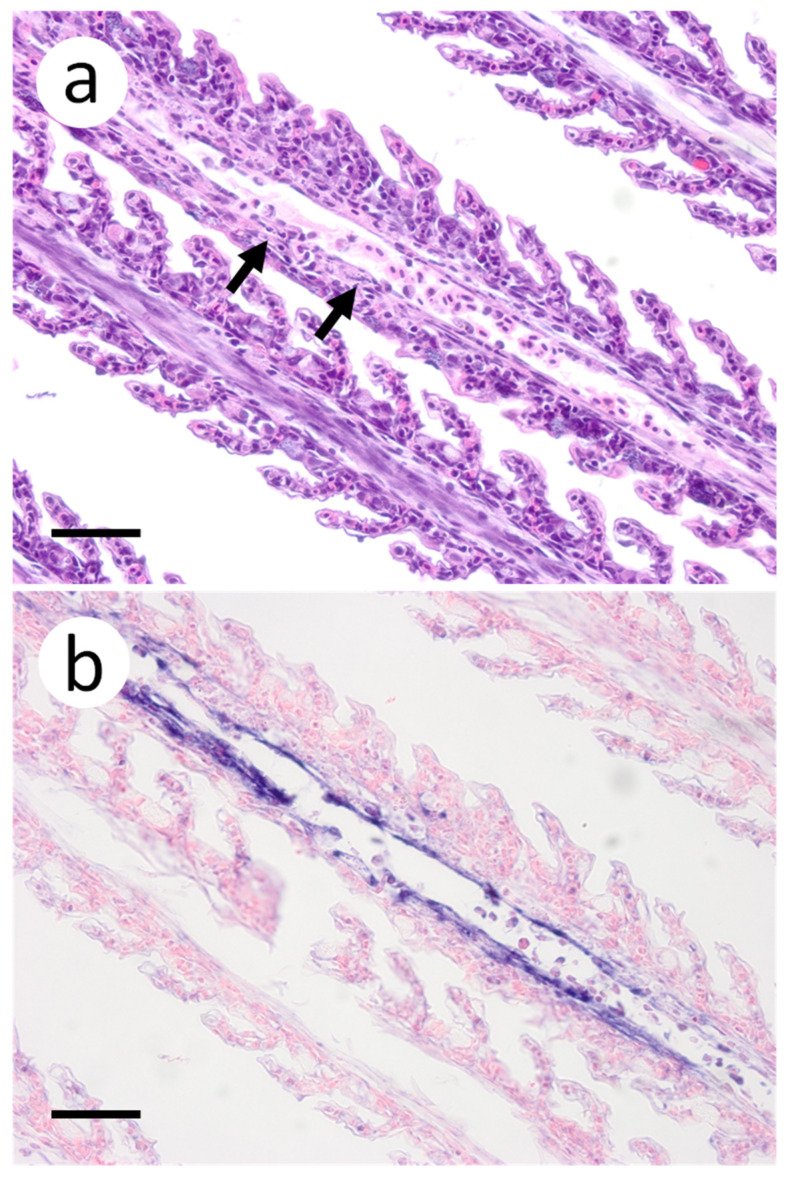
Primary and secondary lamellae of gills obtained from muskellunge (*Esox masquinongy*) exposed at the high dose (1 × 10^6^ plaque forming units ml^−1^), 6 days post-infection. (**a**) Hematoxylin and eosin-stained sections reveal pyknotic nuclei of endothelia lining the primary lamellar space (black arrows). (**b**) In situ hybridization reveals intense riboprobe staining (blue) at the same sight. (Scale bar = 65 µm.)

**Figure 4 animals-12-01624-f004:**
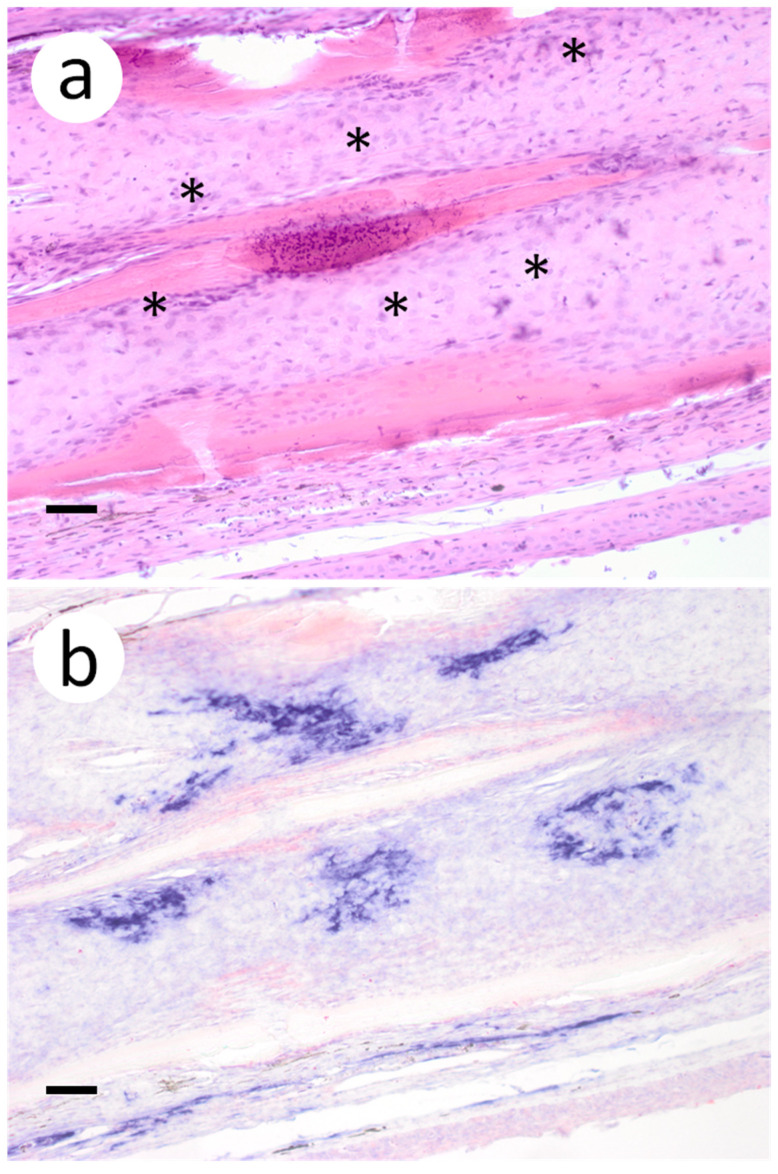
Pectoral fin sections from muskellunge (*Esox masquinongy*) collected from fish exposed at the medium dose (4 × 10^3^ plaque forming units mL^−1^), 6 days post-infection. (**a**) multifocal necrosis of fibroblasts adjacent to fin rays, characterized by pyknotic and fragmented nuclei (asterisks). (**b**) positive riboprobe staining (blue). (Scale bar = 70 µm.)

**Figure 5 animals-12-01624-f005:**
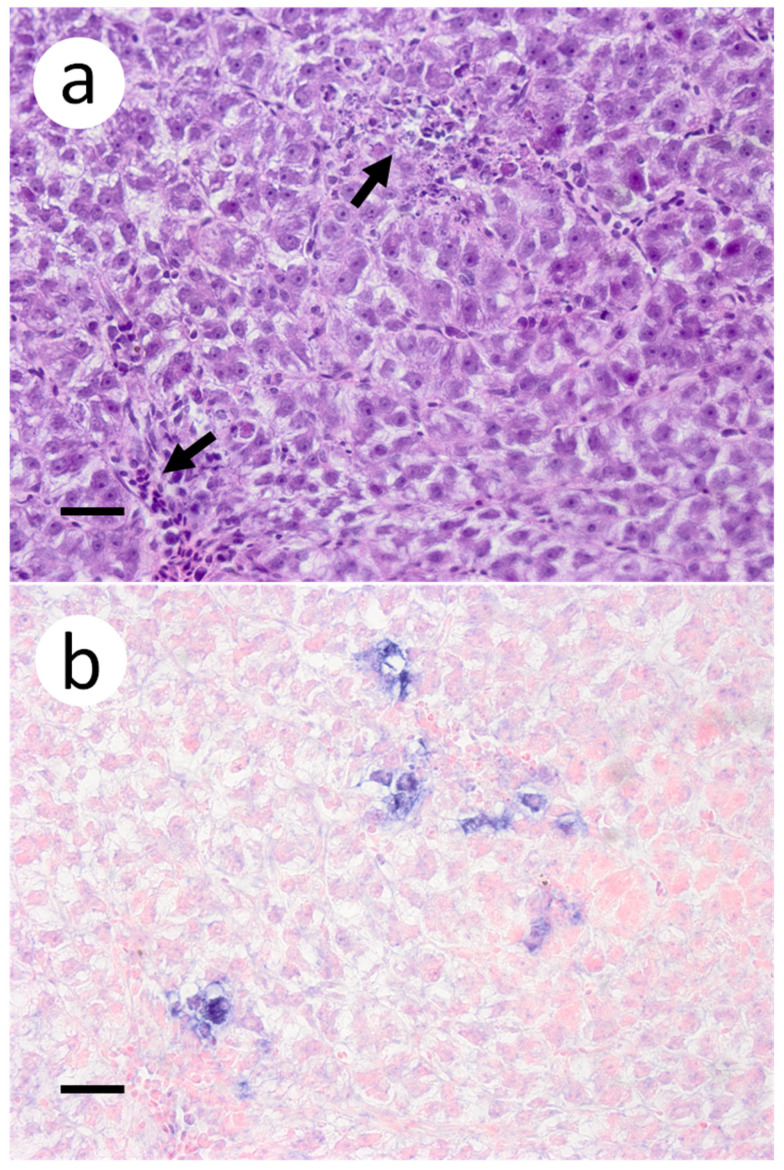
Liver sections from muskellunge (*Esox masquinongy*) exposed at the medium dose (4 × 10^3^ plaque forming units mL^−1^), 28 days post-infection. (**a**) Multifocal hepatocellular necrosis (black arrows). (**b**) Positive riboprobe staining of hepatocytes (blue) (scale bar = 75 µm).

**Figure 6 animals-12-01624-f006:**
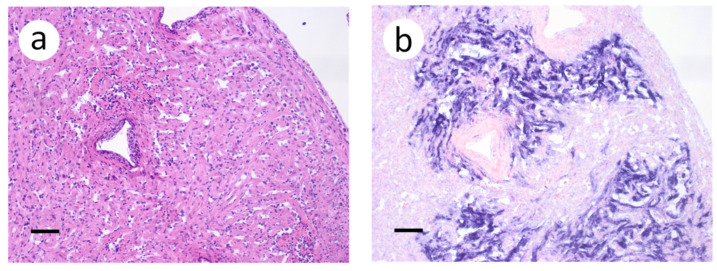
Heart sections from a muskellunge (*Esox masquinongy*) exposed at the medium dose (4 × 10^3^ plaque forming units mL^−1^), 28 days post-infection. (**a**) Severe mononuclear infiltration between necrotic cardiac myocytes. (**b**) Positive riboprobe staining of myocardium. Note the lack of staining in the endo and pericardial layers. (**c**) Focally diffuse cardiac myositis. (**d**) Positive riboprobe staining in cardiac myocytes surrounding the area of inflammation and necrosis (scale bar **a**,**b** = 110 µm; **c**,**d** = 75 µm).

**Figure 7 animals-12-01624-f007:**
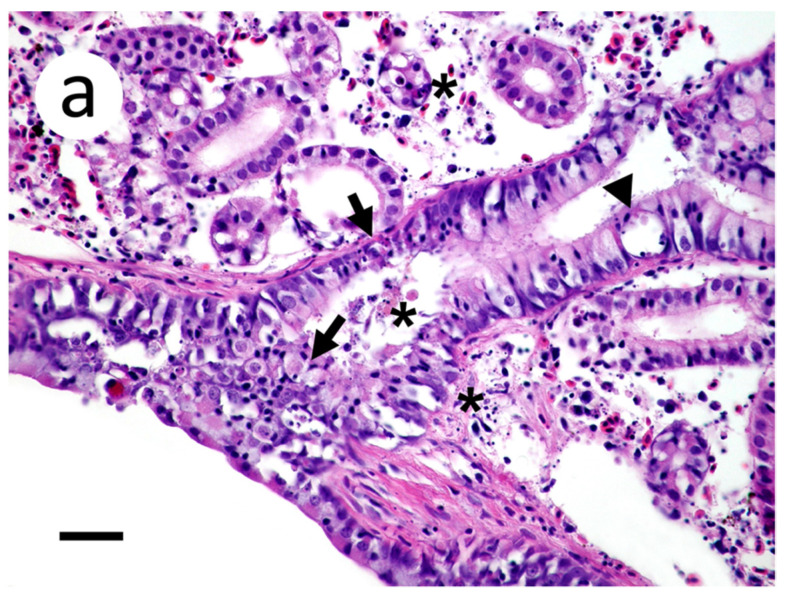
Kidney section of muskellunge (*Esox masquinongy*) exposed at the medium dose (4 × 10^3^ plaque forming units mL^−1^), 6 days post-infection. (**a**) Degenerate renal tubular epithelium are swollen and vacuolated (arrowhead) or shrunken and necrotic with pyknotic nuclei (arrows). There are scattered foci of pyknotic nuclei and cellular debris within renal tubular lumena and interstitium (asterisks). (**b**) Positive riboprobe staining in areas of necrotic cellular debris and admixed with degenerate renal tubular epithelium (scale bar = 60 µm).

**Figure 8 animals-12-01624-f008:**
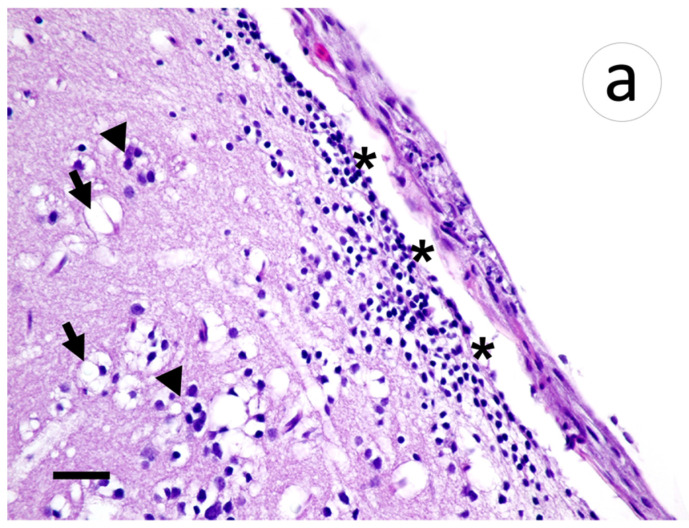
Section of meninges with subjacent gray matter of muskellunge (*Esox masquinongy*) exposed at the medium dose (4 × 10^3^ plaque forming units mL^−1^), 6 days post-infection. (**a**) Spongiosis (arrows), gliosis (arrowheads), and lymphocytic infiltrates within the gray matter and minimally into the meninges (asterisks). (**b**) Positive riboprobe staining in areas of necrosis (scale bar = 60 µm).

**Figure 9 animals-12-01624-f009:**
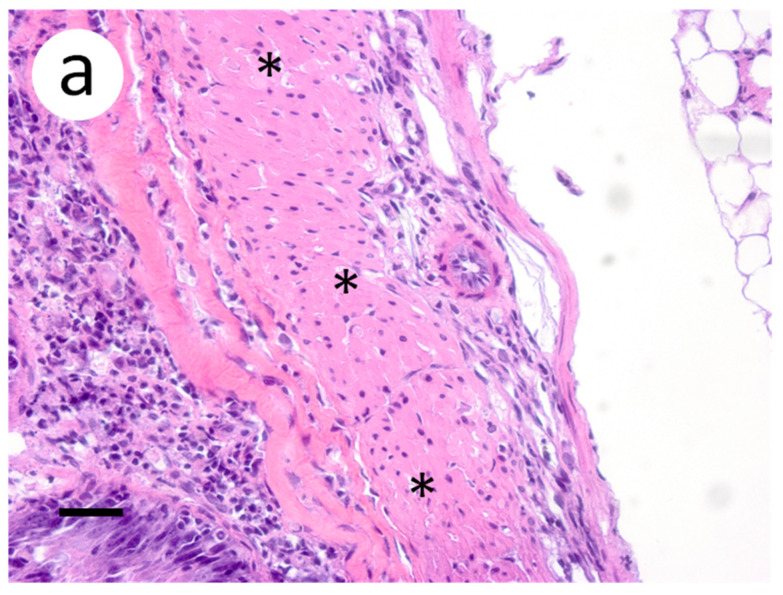
Longitudinal section of the posterior intestine in muskellunge exposed at the medium dose (4 × 10^3^ plaque forming units mL^−1^), 6 days post-infection. (**a**) Single cell pyknosis smooth muscle (asterisk). (**b**) Positive riboprobe staining (blue) in smooth muscle cells intermixed in regions of unaffected regions. (Scale bar = 75 µm.)

**Figure 10 animals-12-01624-f010:**
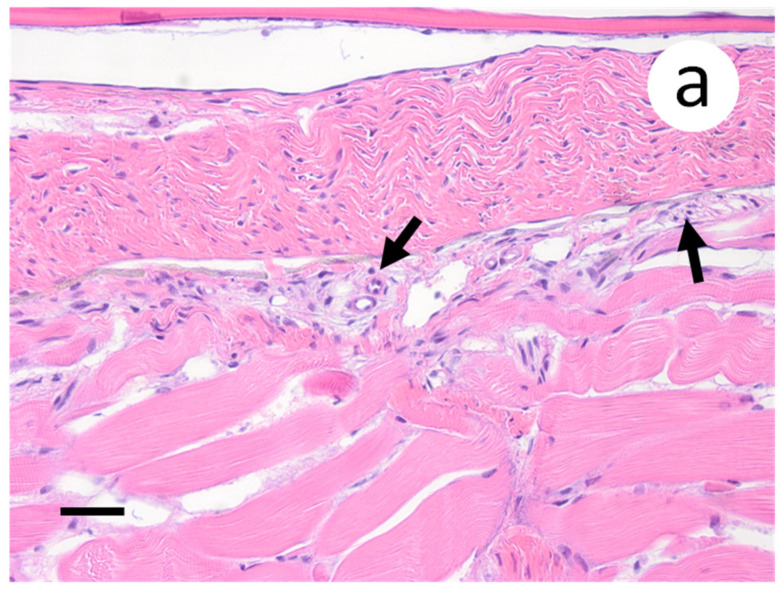
Skin and muscle sections of muskellunge (*Esox masquinongy*) exposed at the medium dose (4 × 10^3^ plaque forming units mL^−1^), 6 days post-infection. (**a**) Mild endothelila degeneration characterized by pyknotic nuclei (black arrows). (**b**) Positive riboprobe staining observed in endothelial cells and at the base of the scales (black arrow) (scale bar = 90 µm).

**Table 1 animals-12-01624-t001:** Distribution of viral hemorrhagic septicemia virus IVb (VHSV-IVb) in muskellunge tissues collected in parallel at predetermined time points in fish challenged at the high dose (1 × 10^5^ PFU mL^−1^). The number of VHSV-IVb positive fish and range of virus load in each tissue is indicated below.

	VHSV Positive Samples (*n*= 4 per Sampling Time)																		Range of Viral Load In Positive Tissues (PFU g^−1^)
	Hours					Days													
Tissue	0 h	6 h	12 h	24 h	36 h	2 d	4 d	6 d	8 d	15 d	22 d	29 d	36 d	43 d	50 d	57 d	64 d	Total	
spleen	0	0	0	0	0	0	0	1	0	1	0	0	0	1	0	1	0	4	1.98 × 10^2^–4.46 × 10^6^
kidney	0	0	0	0	0	0	0	2	1	2	1	0	0	0	0	1	1	8	1.20 × 10^3^–1.56 × 10^7^
heart	0	0	0	0	0	2	0	2	1	2	1	2	1	0	1	2	2	16	2.17 × 10^2^–4.26 × 10^7^
liver	0	0	0	0	0	0	0	3	1	2	3	2	2	0	0	1	2	16	4.51 × 10^2^–5.95 × 10^5^
intestine	0	0	0	0	0	0	0	1	0	2	1	1	2	0	0	0	2	9	1.83 × 10^3^–1.95 × 10^6^
gill	0	0	0	0	0	0	0	2	1	1	2	2	1	0	0	1	3	13	5.29 × 10^2^–9.89 × 10^6^
pectoral fin	0	0	0	0	0	1	0	2	2	2	1	1	0	3	1	1	2	16	1.85 × 10^2^–6.63 × 10^7^
skin/muscle	0	0	0	0	0	0	0	2	2	3	1	2	0	3	0	0	1	14	1.90 × 10^2^–4.81 × 10^5^
Total	-	-	-	-	-	3	-	15	8	15	10	10	6	7	2	7	13	96	

**Table 2 animals-12-01624-t002:** Distribution of viral hemorrhagic septicemia virus IVb (VHSV-IVb) in muskellunge tissues collected in parallel at predetermined time points in fish challenged at the medium dose (4 × 10^3^ PFU mL^−1^). The number of VHSV-IVb positive fish and range of virus load in each tissue is indicated below.

	VHSV Positive Samples (*n*= 4 per Sampling Time)																		Range of Viral Load In Positive Tissues (PFU g^−1^)
	Hours					Days													
Tissue	0 h	6 h	12 h	24 h	36 h	2 d	4 d	6 d	8 d	15 d	22 d	29 d	36 d	43 d	50 d	57 d	64 d	Total	
spleen	0	0	0	0	0	0	0	1	0	1	0	2	0	2	0	0	1	7	1.85 × 10^4^–1.30 × 10^8^
kidney	0	0	0	0	0	0	0	3	0	1	1	2	0	2	0	0	1	10	5.96 × 10^2^–2.63 × 10^7^
heart	0	0	0	0	0	0	0	3	0	1	2	3	0	3	0	0	3	15	5.72 × 10^3^–5.93 × 10^8^
liver	0	0	0	0	0	0	0	2	0	2	1	2	0	2	0	0	1	10	8.21 × 10^2^–5.20 × 10^8^
intestine	0	0	0	0	0	0	0	2	0	1	2	4	0	2	0	0	0	11	1.23 × 10^3^–3.17 × 10^7^
gill	0	0	0	0	0	0	0	3	0	2	2	3	0	2	0	1	1	14	1.49 × 10^3^–5.04 × 10^8^
pectoral fin	0	0	0	1	0	0	0	3	0	1	0	3	0	2	1	0	1	12	5.15 × 10^3^–8.16 × 10^7^
skin/muscle	0	0	0	0	0	0	0	3	1	3	0	3	0	3	1	0	1	15	1.31 × 10^2^–2.82 × 10^7^
Total	-	-	-	1	-	-	-	20	1	12	8	22	-	18	2	1	9	94	

**Table 3 animals-12-01624-t003:** Distribution of viral hemorrhagic septicemia virus IVb (VHSV-IVb) in muskellunge tissues collected in parallel at predetermined time points in fish challenged at the low dose (100 PFU mL^−1^). The number of VHSV-IVb positive fish and range of virus load in each tissue is indicated below.

	VHSV Positive Samples (*n*= 4 per Sampling Time)																		Range of Viral Load In Positive Tissues (PFU g^−1^)
	Hours					Days													
Tissue	0 h	6 h	12 h	24 h	36 h	2 d	4 d	6 d	8 d	15 d	22 d	29 d	36 d	43 d	50 d	57 d	64 d	Total	
spleen	0	0	0	0	0	0	0	0	0	0	0	0	0	0	0	0	0	0	-
kidney	0	0	0	0	0	0	0	0	0	0	0	0	0	0	0	0	0	0	-
heart	0	0	0	0	1	0	0	0	0	0	0	0	0	0	0	0	0	1	1.58 × 10^2^
liver	0	0	0	1	0	0	1	0	0	0	1	0	0	0	0	0	0	3	1.08 × 10^3^–2.76 × 10^3^
intestine	0	0	0	0	0	0	0	0	0	0	1	1	0	1	0	0	0	3	1.78 × 10^2^–1.97 × 10^4^
gill	0	0	0	0	0	0	0	0	0	1	0	0	0	0	0	0	0	1	2.74 × 10^3^
pectoral fin	0	0	0	0	0	1	0	0	0	0	0	0	0	0	0	0	0	1	4.47 × 10^3^
skin/muscle	0	0	0	0	0	0	0	0	0	0	0	0	0	0	0	0	0	0	-
Total	-	-	-	1	1	1	1	-	-	1	2	1	-	1	-	-	-	9	

**Table 4 animals-12-01624-t004:** Estimations of β values and odds ratio (OR) of the sampled tissues. The tissues are listed in order of probability that viral hemorrhagic septicemia virus genotype IVb can be reisolated versus the kidney.

Tissues	Estimate β	OR of Tissue vs. Kidney (e^β^)	95% Confidence Limit for OR	
Liver	0.7847	2.1917	1.3860	3.4660
Heart	0.6025	1.8267	1.2411	2.6885
Skin/muscle	0.5197	1.6815	1.1073	2.5536
Gill	0.3586	1.4313	1.0096	2.0293
Pectoral fin	0.3056	1.3574	0.9945	1.8528
Intestine	0.2151	1.2400	0.7857	1.9570
Kidney	0.0000	1.0000	-	-
Spleen	−0.3215	0.7251	0.4377	1.2009

**Table 5 animals-12-01624-t005:** P value of paired t-test results of the overall mean viral load from the eight tissues. The significance level (α) was modified through the Bonferroni correction by dividing the α level (0.10) by the total number of pairs (28). Significant pairings are bolded.

	Kidney	Heart	Liver	Intestine	Gill	Pectoral Fin	Skin/Muscle
spleen	0.2339	<0.0001	<0.0001	0.0638	0.0030	0.0014	0.0012
kidney	-	0.0003	0.0005	0.3489	0.0150	0.0023	0.0150
heart	-	-	0.6331	0.0138	0.1435	0.3299	0.4112
liver	-	-	-	0.0042	0.0673	0.0673	0.2354
intestine	-	-	-	-	0.2838	0.2484	0.2075
gill	-	-	-	-	-	0.7493	0.7340
pectoral fin	-	-	-	-	-	-	0.9424

Bonferroni correction *p* value: 0.10/28 = 0.00357, significance = value(s) < 0.00357.

## Data Availability

The data presented in this study are available on request from the corresponding author.

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
