# Peer review of "Tissue Distribution of the Piscine Novirhabdovirus Genotype IVb in Muskellunge (Esox masquinongy)"

_animals, 2022, doi:10.3390/ani12131624_

Round 1

Reviewer 1 Report

            Viral hemorrhagic septicemia virus (piscine novirhabdovirus, VHSV) is a widely distributed pathogens of fish. VHSV is capable of infecting a large number of fish species. In the present paper experimental infection of muskellunge with VHSV genotype IV was analyzed, with particular focus on the tissue distribution of the virus along time. This is an important topic for diasease diagnosis. Other reports of VHSV infection in esocidae fish have been published before. Nevertheless, the present study appears to be very complete, with up to eight different tissues/organs analyzed. It points out how different can be muskellunge infection from rainbow trout infection with VHSV. Conclusions are supported by plaque assay, PCR and in situ hybridization data, and imply useful tips for non-lethal sampling for diagnosis of VHSV disease.

Specific issues

I believe some further information could be in the manuscript:

1.- Were the VHSV-infected fish experiencing mortalities and/or clinical signs? If so, at what dose of virus?

2.- Which VHSV dose (high, medium, low) would better resemble the natural infection of wild muskellunge?

3.- Were serum samples tested for anti-VHSV antibodies at any time point after infection?

4.- In Table 1, at the medium dose challenge: Is there any explanation for the zero virus isolations at 36 dpi?

References:

The following paper may be relevant to the discussion: Oidtman, B., Joiner, C., Stone, D., Dodge, M., Reese, R.A. and Dixon, P. 2011. Viral load of various tissues of rainbow trout challenged with viral haemorrhagic septicaemia virus at various stages of disease. Dis. Aquat. Org.

 One early work where VHSV was found in heart on infected fish tha might be useful: Miller TA, Rapp J, Wastlhuber U, Hoffmann RW, Enzmann PJ. Rapid and sensitive reverse transcriptase-polymerase chain reaction based detection and differential diagnosis of fish pathogenic rhabdoviruses in organ samples and cultured cells. Dis Aquat Organ. 1998 Sep 11;34(1):13-20. doi: 10.3354/dao034013.

Reviewer 2 Report

ABSTRACT

L26 Please provide Latin name for muskellunge 

L27 Please provide a clear information about high, medium, and low doses. 

INTRODUCTION

L49 “historically known” – What about recent outbreaks? 

L58 Please specify which samples do you mean.

L65 What’s the detection limit using the cell culture?

L76 Please a dot “.” after word “turbot”.

L86 Lake herring is not a species within Salmonidae family. Please rephrase this sentence considering usage of word “both”. 

L89-91 Information about ISH should be provided in the fragment where the other methods are discussed. Do not mix different aspects (strains, methodology, etc.) the introduction. 

L93 Please clarify both objectives of your study. 

MATERIAL AND METHODS

L104 “C” is not a valid unit for temperature level. 

L110/134 Please delete space between the “25” and the “°C”. `Check also other instances in the manuscript.

L127 and 128 Do not use first capital letters for the names of the antibiotics. 

L136/137 Please provide details for water flow, temperature and DO level.

L138 “ad lib” is not a correct form. Please use “ad libitum

L167 Please provide details for the dilution medium.

L168 at what speed? 

L177/178 Information on negative controls is needed here.
